# Semi-Precast Segmental Bridge Construction Method: Experimental Investigation on the Shear Transfer in Longitudinal and Transverse Direction

**Stephan Johann Fasching** *,†  , **Tobias Huber** †, **Michael Rath and Johann Kollegger**

Institute for Structural Engineering, TU Wien, Karlsplatz 13/212-2, 1040 Vienna, Austria;
tobi.huber@tuwien.ac.at (T.H.); michael.rath@tuwien.ac.at (M.R.); johann.kollegger@tuwien.ac.at (J.K.)
* Correspondence: Stephan.Fasching@tuwien.ac.at; Tel.: +43-1-58801-21259
† These authors contributed equally to this work.

**Abstract:** Large span concrete bridges with a box-shaped girder are usually built from prefabricated concrete segments or by in-situ casting of the concrete on a scaffolding system. Both technologies have their advantages and drawbacks. Recently a new approach to the construction of such bridges which combines the advantages of both existing solutions was proposed at the TU Wien. This method uses the standard precast segmental erection methods with their high construction speed, but divides the segments into easily transportable pre-fabricated thin-walled elements to create new, lighter versions of the segments. Following the installation of these lightweight segments, they are strengthened with additional concrete in their final position in the superstructure. This paper focuses on the transmission of shear forces during construction stages. Firstly, on the level of individual segments, where rigid cross-frames are necessary to guarantee the stability of the segments and secondly, on the level of a bridge girder built from such segments, where new joint types must be developed to ensure the force transfer between the segments. Different options for the formation of cross-frames as well as shear tests on double walls and concrete panels with steel girders are shown. In this experimental series, different shear transmitting elements were compared to each other and to calculations with non-linear finite element analysis, showing that all the investigated solutions are suitable for use in thin-walled bridge segments. Several methods, including a new concept for joining thin-walled pre-fabricated elements, are described for the joints between the segments. Push-off tests with a constant lateral force were carried out to assess the shear strength and deformation behaviour. The main parameters were the joint type (wet joints: plain, grooved, keyed; dry joints), the grout type, and the lateral force level. The test results are presented and the structural behaviour is further analysed using non-linear finite-element simulations.

**Keywords:** thin-walled pre-fabricated elements; post-tensioning; prefabrication; joints; cross-frames; bridge construction; semi-precast-segmental bridges; unfilled double walls; segmental bridges; push-off test; shear testing; shear; experiments

## 1. Introduction

Bridges are an integral part of the existing central European infrastructure. Since most of the traffic routes in Europe already exist and the loads to be considered are rising steadily [1], in addition to maintaining and assessing existing bridges, the replacement of old structures is often unavoidable. Minimising construction time, traffic disruption and fuel wastage, as well as emotional distress, are of particular importance for such replacement projects [2].

The Institute for Structural Engineering at the TU Wien has developed new construction methods for the use of semi-precast concrete elements in bridge construction. These technologies can be used for T-beams, which are applied on short to medium spans [3–5], for deck slabs on bridges [6], and for girders with a box shaped cross-section [7–9], which

are used for large multi-span bridges [10,11]. By employing these technologies, not only construction time, but also construction cost can be reduced, [3,6], while the combination of prefabricated parts and in-situ concrete creates a durable, monolithic superstructure [12].

In [7] a new approach to the construction of box-shaped segments from thin-walled concrete panels (70 to 100 mm), strengthened by steel girders (Figure 1a, as well as the construction of a prototype, shown in Figure 1b was presented. The idea of this construction method is to install such lightweight segments in a bridge girder with existing segmental bridge construction methods, for example, incremental launching or the balanced cantilever method as described in [10,11,13]. This girder acts as formwork and falsework for further in-situ concrete, finally creating a monolithic superstructure. The big advantage in comparison to segments with final dimensions is the reduced weight, resulting in lower stresses during construction as well as easier handling and transport. In comparison to in-situ casting of concrete, it offers more rapid construction and the shifting of working steps from the site to a factory.

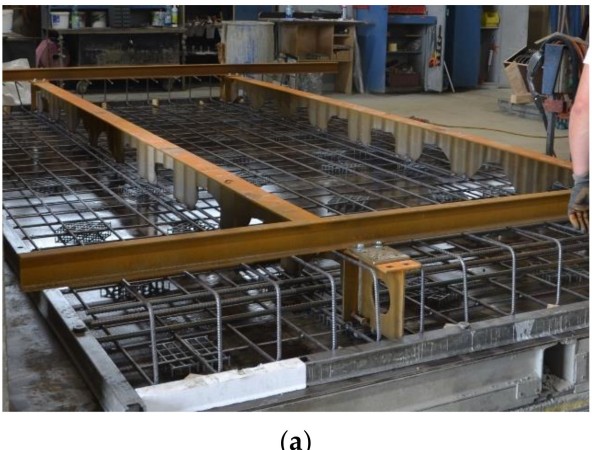
(**a**)

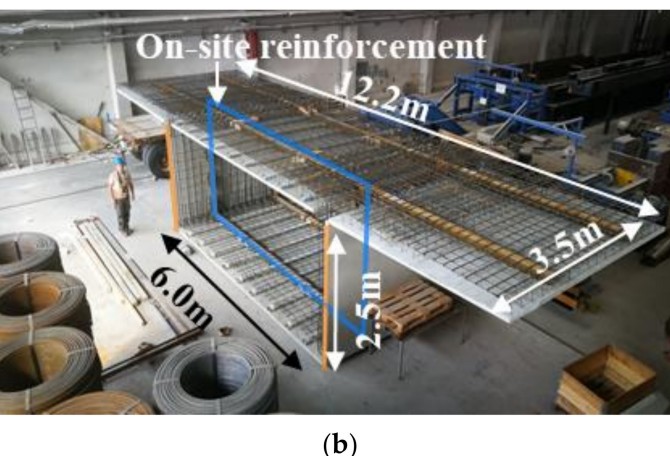
(**b**)

**Figure 1.** Construction of a lightweight bridge segment using thin-walled prefabricated elements with 70 mm thickness (**a**) corrugated steel web girders on a casting bed (**b**) prototype bridge segment with highlighted cross-frame in blue and highlighted surface for shear transmission in orange (taken from [7]).

Figure 2c shows the application of such lightweight segments with cross-frames to the balanced cantilever method. After the installation of a certain number of segments, the shear force distribution due to self-weight along the bridge girder, as shown in Figure 2a, can be observed. This shear force needs to be transferred across grouted joints (highlighted in orange in Figures 1b and 2a. In the next step, after assembling lightweight segments to a girder, in-situ concrete is added to strengthen the thin-walled structure in a targeted manner. In the example of the balanced cantilever bridge in Figure 2, the bottom slab is strengthened first above the pier (Figure 2b,c), causing the shear force distribution in the cross-frames of the segments displayed in Figure 2b.

This paper focuses on the transmission of shear forces during the construction of a bridge with this new method: on the one hand, in the transversal direction at the level of one cross frame in a segment, and on the other hand at the level of a bridge girder in the longitudinal direction, across the grouted joints between segments.

### 1.1. Shear Transfer in Transverse Direction

In the transversal direction, a box-shaped girder is a quadrangle with two cantilevers (Figure 2b). Therefore, the corners of this box must be rigid to guarantee a non-kinematic system during all construction stages. In addition to the necessity for rigid corners, the concrete panels and the steel beams with their corrugated webs (Figure 1a) must act as compound elements, creating cross-frames in the segments. For this reason, the corrugated

webs must be capable of transferring shear forces between the concrete panel and the flange of the steel girder.

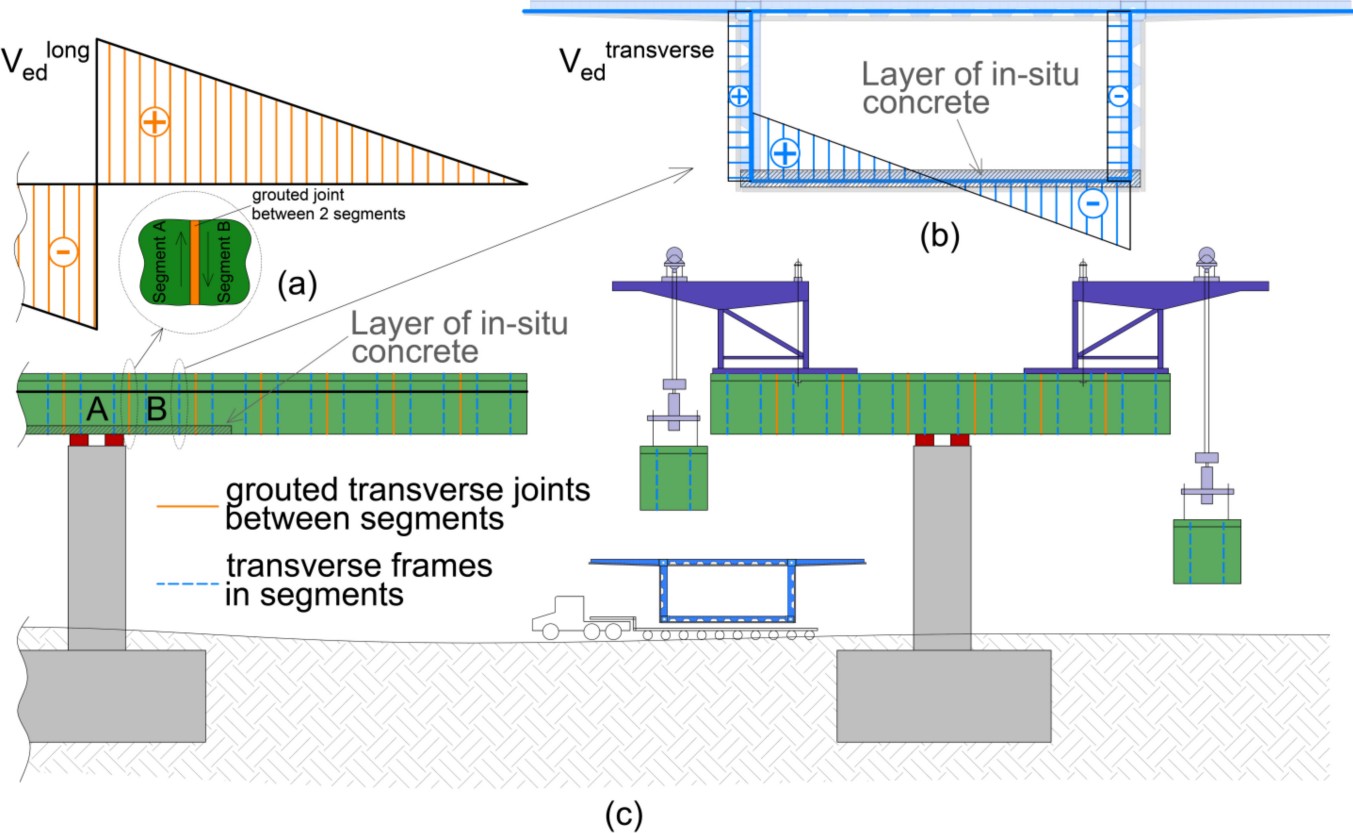

**Figure 2.** Application of thin-walled pre-fabricated elements for the construction of a balanced cantilever bridge with lifting frames: (**a**) distribution of shear forces in longitudinal direction due to self-weight; (**b**) distribution of shear forces in a cross-frame due to adding in-situ concrete to the bottom slab (**c**) illustration of the construction process (drawing based on [7]).

As additional formwork placed on the inner sides of the segment's webs (see Figure 1b) would be necessary for their completion, it was decided to investigate the suitability of double wall elements for the vertical webs of the lightweight segments. The design idea for this version was to connect the two shells of the double walls with steel elements which would be strong enough to allow for the two individual panels to act as one bending beam.

While investigations into corrugated webs in steel–concrete-compound structures have been conducted [14–18], none exist for SIN-shaped steel webs and the chosen type of connection between steel web and concrete. To the best of our knowledge, there have been no investigations into the shear behaviour of unfilled double wall elements.

To investigate the shear capacity of thin-walled concrete elements with steel girders as well as unfilled double wall elements with different types of connecting elements made of steel, we investigated 10 test specimens (six double walls and four panels with steel girders) experimentally and with non-linear finite element analysis.

### 1.2. Shear Transfer in Longitudinal Direction, between Segments

The construction of segmental bridges was made possible by Eugène Freysinnet's invention of post-tensioning in the 1930s [19]. He constructed the first post-tensioned segmental bridge in 1946 [20]. The prestressing force provides the necessary friction to enable the shear forces to be transferred between individual segments. In particular, match-cast technology has proven to be especially efficient in the construction of segmental bridges, with its first application by Ulrich Finsterwalder in Germany in 1951. Jean Mueller,

a former employee of Freysinnet, further developed this method and built the first two bridges with match-cast dry joints and epoxy coated joints around 1960 [20]. Nowadays, the joints of segmental concrete box-girder bridge segments usually have a keyed surface and are still realised with or without epoxy. In our proposed construction method, the division of bridge segments into smaller elements introduces special requirements for in-between joints in respect of tolerances and the subsequent completion of the cross-section.

When connecting bridge segments built with thin-walled elements, a continuous shear transfer between the elements during the construction stage is facilitated by in-place cast mortar joints (orange in Figure 2). The continuous joints also enable compensation of tolerances and the transfer of the post-tensioning forces. The sealed shell also serves as the formwork for the completion of the cross-section.

In this paper, the shear transfer by post-tensioned grouted joints is investigated based on data reported in [21]. In the course of the research, a series of push-off tests with a constant lateral force was conducted to assess the shear strength and deformation behaviour. The main parameters were the joint type (wet joints: plain, grooved, keyed; dry joints), the grout type, and the lateral force level. Below, the full test results are presented, and the structural behavior is further analysed by non-linear finite element simulations.

## 2. Shear Tests on Cross-Frames in Lightweight Segments

### 2.1. Materials and Methods

Ten specimens with different shear transmitting elements were produced and subjected to shear tests under static loading in the laboratories of the TU Wien. With two connecting elements per specimen, a total number of 20 connecting elements were tested. Six of the ten specimens were double wall elements with a thickness of 396 mm, the remaining four had thin-walled concrete panels had attached steel girders with corrugated webs, with an overall height of 361 mm, as shown in the prototype-segment in Figure 1a.

#### 2.1.1. Specimens

Two test specimens of each design, as shown in Figure 3, were produced. The base element of each specimen was a 0.5 m by 0.5 m concrete panel with a thickness of 70 mm. This concrete panel was either connected by the steel elements to be tested to another concrete panel, creating a double wall (Figure 3a–c), or had two attached steel beams with corrugated webs, creating the steel-concrete compound elements (Figure 3d–f), as used for the prototype segment (Figure 1).

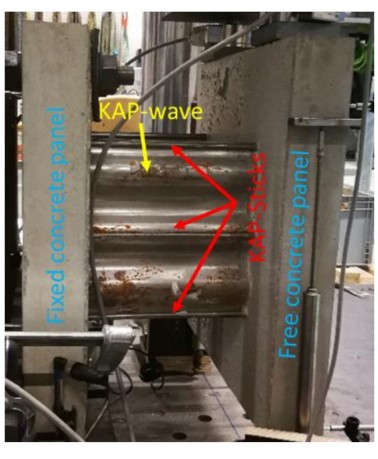
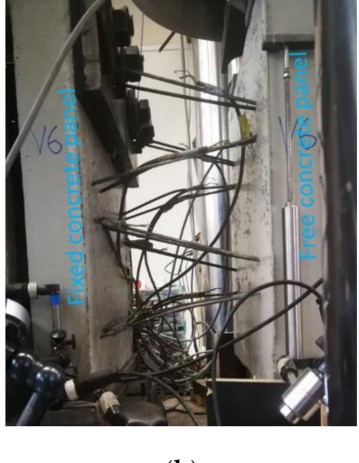
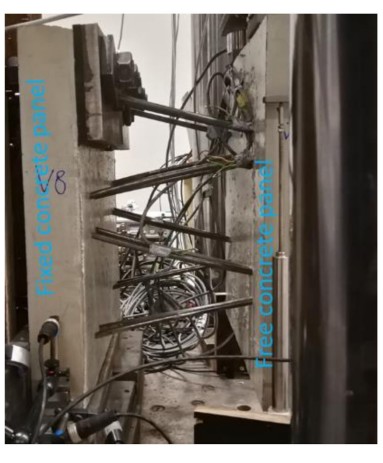

| (**a**) | (**b**) | (**c**) |

**Figure 3.** *Cont.*

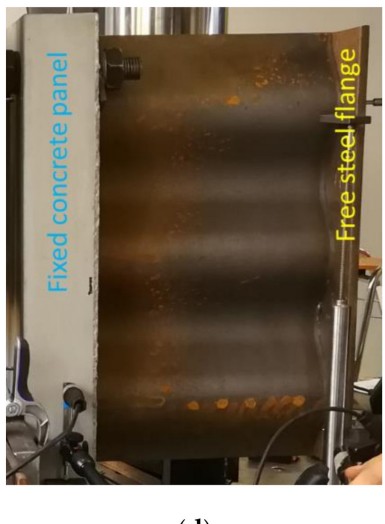

(**d**)

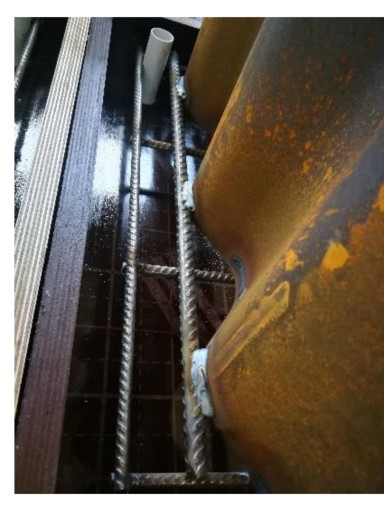

(**e**)

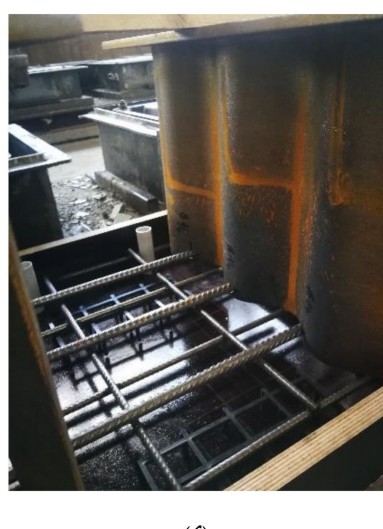

(**f**)

**Figure 3.** Test specimens: (**a**) Double wall with steel connectors; (**b**) Double wall with lattice girders, Ø5 mm diagonals; (**c**) Double wall element with lattice girders, Ø9 mm diagonals; (**d**) Thin-walled concrete panel with corrugated web SIN-T-beam; (**e**) welded bars in longitudinal direction for steel-concrete connection (**f**) transverse bars penetrating the beam for steel-concrete connection.

The investigated parameters in the tests were the shear transmitting components, which were positioned either symmetrically between the two concrete panels or on top of one concrete panel. The different layouts were:

- Double wall with KAP-steel-connectors, consisting of a 0.63 mm thick steel wave and three so called KAP-sticks (Ø6 mm) made from stainless steel [22] (Figure 3a).
- Double wall element with standard lattice girders, Ø5 mm diagonals (type E in [23,24]) (Figure 3b).
- Double wall element with stronger lattice girders, Ø9 mm diagonals (type EV in [23,24]) (Figure 3c).
- Thin panel with corrugated web SIN-T-beam [25,26] and welded longitudinal reinforcement bars Ø12 mm (Figure 3d,e).
- Thin panel with corrugated web SIN-T-beam [25,26] and transverse reinforcement bars Ø12 mm put through holes Ø14 mm (Figure 3d,f).

### 2.1.2. Test Setup, Loading and Measuring Setup

The test setup of the double wall specimen with lattice girders is illustrated from the side and front in Figure 4. This setup is similar to the one in [14], but instead of testing 2 specimens at the same time only one was investigated in a single test. During the displacement-controlled static loading tests, the concrete panel was supported in the horizontal direction by a steel construction using threaded rods and steel panels. In the vertical direction steel panels, lying directly on the ground panel of the testing facility supported the specimens.

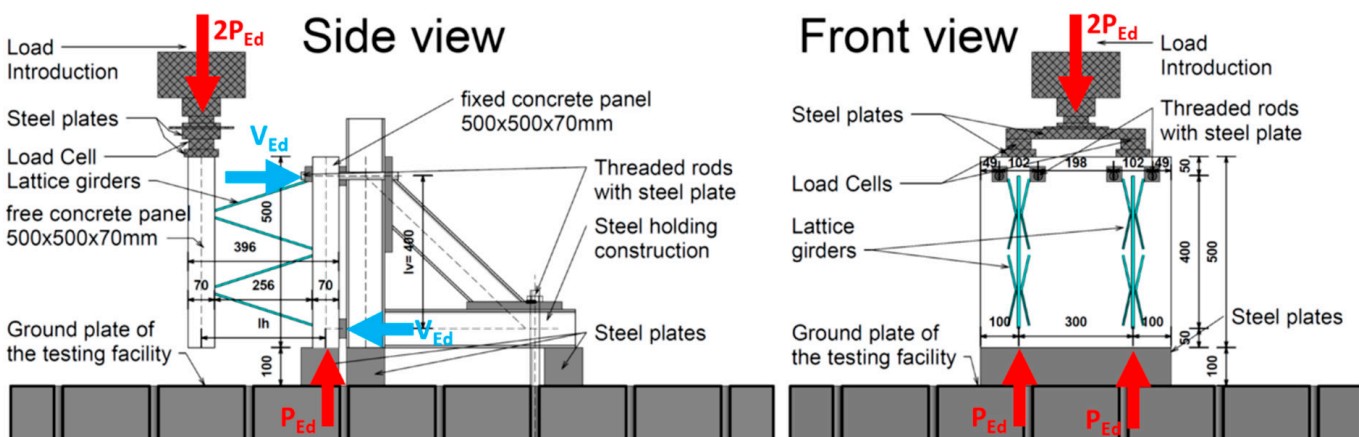

**Figure 4.** Test setup in side-view and front-view (free panel not displayed in front-view), units in mm.

During the tests, the magnitude of the load $P_{Ed}$ applied to the free-floating concrete panel was measured with two load cells (Figure 4) and the vertical and horizontal displacements of both concrete panels with 15 LVDTs in total. The relative displacement between the two shells was calculated from the measured displacements. In addition, the strains on the steel diagonals between the two panels were recorded using strain gauges. In the case of the thin-walled specimens with an attached steel beam, the deformations of the steel flange and the concrete panel were recorded.

The measuring setup was not modified for any of the tested specimens, while the test setup was changed for two of the four specimens with steel beams. Two were tested with the concrete panel fixed and the steel beam free-floating (Figure 3d), which resulted in a pull-out of the steel girders from the concrete panel. Therefore, the setup was changed for the remaining two specimens, which were tested twice: first with a horizontal support at the upper part of the steel beam (described in detail in Section 2.3.2), and second with the steel beam fixed and the concrete panel free-floating (described in detail in Section 2.3.2).

### 2.1.3. FE-Analysis

All specimens were modelled three dimensionally with the calculation parameters shown in Table 1, which also shows the parameters for the modelled push-off tests described in Section 3 below. To reduce the computational effort, only one connecting element was modelled for each investigated specimen (Figure 5). This resulted in concrete panels with a size of $200 \times 500 \times 70$ mm and one connecting element between two panels, or one embedded SIN-beam, respectively.

Figure 5a shows the supports, modelled as fixed boundary conditions in the respective direction. Additionally, a symmetry boundary-condition was implemented at one side of the free-floating panel or on the steel flange. The load was introduced as a displacement of 10 mm via a steel panel on top of the specimens, as in the real tests.

The analysis was performed with 3 calculation steps:

1. Boundary conditions: vertical support, horizontal support, symmetry
2. Self-weight
3. Loading phase: application of 10 mm deformation

At the lattice girders diagonals, deformations in the longitudinal direction of the girders towards the lower belt (see Figure 5b and Table 2) with different magnitudes were observed. The influence of the magnitude of imperfections was studied by running several FE-simulations with magnitudes of imperfections from 0 mm (perfect geometry) to 6 mm, as shown in Figure 5b.

**Table 1.** Parameters used for the FEA-simulations in the software Abaqus for cross-frames and push-off tests on joints.

| Parameter | | Cross Frames in Segments | | | | | Push-Off Tests on Joints between Segments (Section 3) |
|---|---|---|---|---|---|---|---|
| | | Lattice Girder 5 mm | Lattice Girder 9 mm | Kap Wave | SIN-Welded | SIN-Penetrated | |
| **Calculation Procedure** | | Abaqus/Dynamic, Implicit (Application: Quasi Static) | | | | | |
| Elements (Element size) | C3D8R | - | - | Kap Sticks (2 mm) | SIN beam flange (5 mm), bond reinforcement (5 mm), SIN beam part embedded in concrete (5 mm) | | V1–3 Concrete-specimen (30 mm–2.5 mm) V2–3 Grout (2.5 mm) V4 Grout (3 mm) |
| | C3D10 | Concrete panels (20 mm) | | | | | V4 Concrete-specimen (30 mm-3 mm) |
| | | Lattice girder (2.7 mm) | Lattice girder (5 mm) | - | | | |
| | T3D2 | panel reinforcement (20 mm) | | | | | specimen reinforcement (20 mm) |
| | S4R | - | - | Kap Wave (2.5 mm) | Corrugated Web (10 mm) | | - |
| Interaction | Steel-Concrete Concrete-Grout | Tie | | | | normal: hard contact, transverse: penalty (μ = 0.3) | Concrete-Grout: normal: hard contact, transverse: penalty (μ = 0.8) V2: friction and cohesion |
| | Reinforcement-concrete | embedded | | | | | |
| | Corrugated web-bond reinforcement | - | | | Tie | normal: hard contact, transverse: penalty (μ = 0.3) | - |
| | Shell to solid coupling | - | | | free to embedded part of corrugated web | | - |
| **Solution technique** | | Full newton | | | | | |
| Material model | Concrete | Concrete damaged plasticity (dilation angle = 35°; eccentricity = 0.1; $f_{b0}/f_{c0}$ = 1.16; K = 0.667; Viscosity parameter = 0) | | | | | |
| | Elastic-plastic | panel reinforcement | | | | | specimen reinforcement |
| | | Lattice girder | | Kap-Wave, Kap-Sticks | SIN Beam | | |

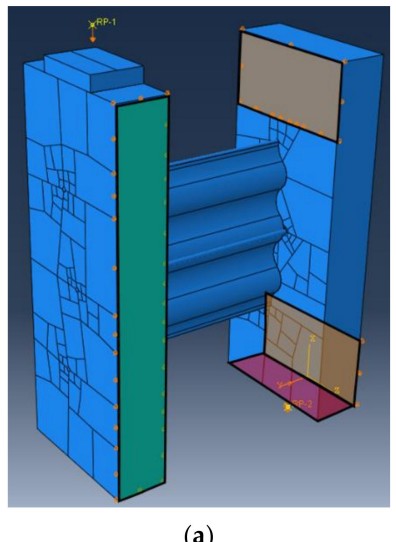
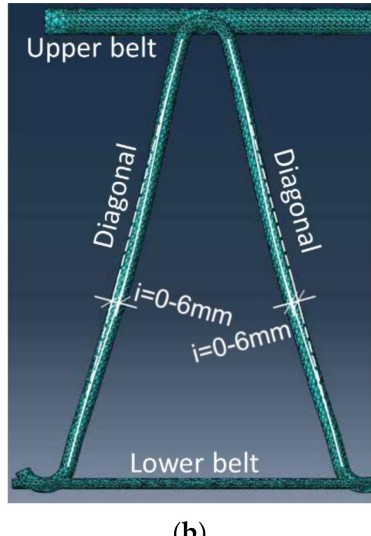
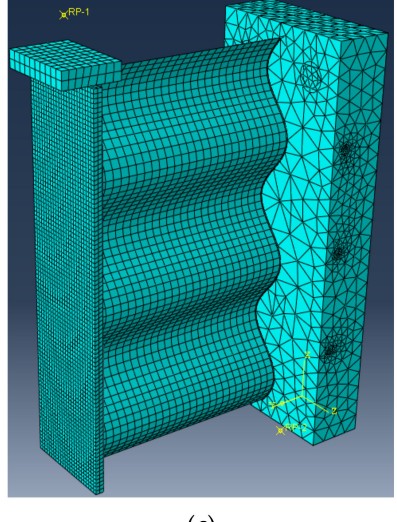

(**a**)    (**b**)    (**c**)

**Figure 5.** Modelling of tested specimen in Abaqus: (**a**) Double wall with KAP-steel-connector, showing the boundary conditions: vertical support (red), horizontal support (orange), symmetry (green), deformation at the Reference point (**b**) 9 mm Lattice girder with pre-deformed geometry, deformation towards the lower belt (**c**) Concrete panel with SIN-beam after mesh generation.

**Table 2.** Measured imperfections at the lattice girders.

| Lattice Girder Type | Imperfection Longitudinal | Imperfection Transverse |
|:---:|:---:|:---:|
| Ø5 mm | 0.5 mm to 3.0 mm | 0.5 mm to 2.0 mm |
| Ø9 mm | 1.5 mm to 4.5 mm | 0.0 mm to 1.0 mm |

Figure 5c shows a model of a concrete panel with an attached SIN-beam without imperfections after the generation of the mesh with the elements as in Table 1.

The material characteristics were obtained in laboratory tests for the concrete and the lattice girders diagonals and were received from the producer of the SIN-beams and the KAP-steel-connectors. The material models used in the FE-software were calibrated by running compression and tensile tests in the software. For the rebar, standard values were considered.

Concrete damaged plasticity model [27]:

For compressive behaviour, the model according to Sargin [28] was used without damage. The tensile behaviour was modelled according to Hillerborg [29], with damage according to [30] and a maximum damage of 95% for numerical reasons. The material parameters are shown in Table 3, while the strains were considered according to Eurocode 2 [31] for concrete class C70/85.

Elastic-plastic steel model

**Table 3.** Abaqus material parameters for concrete for tests on cross frames and push-off joints.

| Specimen | | $\rho$ | E | $\nu$ | $f_{cm}$ | $f_{ctm}$ |
|:---:|:---:|:---:|:---:|:---:|:---:|:---:|
| | | g/cm$^3$ | MPa | - | MPa | MPa |
| Cross-frames | Concrete panels | 2.30 | 41,000 | 0.20 | 75.00 | 4.50 |
| Push-off tests on joints | Grout 1 | 2.24 | 41,000 | 0.20 | 75.40 | 5.13 |
| | Grout 2 | 2.26 | 41,000 | 0.20 | 84.00 | 3.00 |
| | Concrete specimen | 2.33 | 41,000 | 0.20 | 81.40 | 4.10 |

In tensile tests, nominal stresses (see Table 4) were obtained, which were transformed into true stresses according to [27,32] and used in the software. The stress–strain relationships are shown in Figure 6. For the reinforcement, a simple model was chosen since it had no influence on the results.

**Table 4.** Material parameters (nominal stresses) for steel (reinforcement used for cross-frames and push-off tests).

| | $\rho$ | E | $\nu$ | $f_y$ | $f_u$ | $\varepsilon_u$ |
|:---:|:---:|:---:|:---:|:---:|:---:|:---:|
| | g/cm$^3$ | MPa | - | MPa | MPa | (%) |
| Lattice girder 5 mm | 7.85 | 212,000 | 0.3 | 573 | 630 | 6.00 |
| Lattice girder 9 mm | 7.85 | 204,200 | 0.3 | 538 | 607 | 11.00 |
| KAP-Stick | 7.85 | 200,000 | 0.3 | 350 | 450 | 25.00 |
| KAP-wave | 7.85 | 208,000 | 0.3 | 208 | 312 | 42.00 |
| SIN-Beam | 7.85 | 210,000 | 0.3 | 320 | 440 | 27.50 |
| Reinforcement | 7.85 | 200,000 | 0.3 | 560 | 600 | 30.00 |

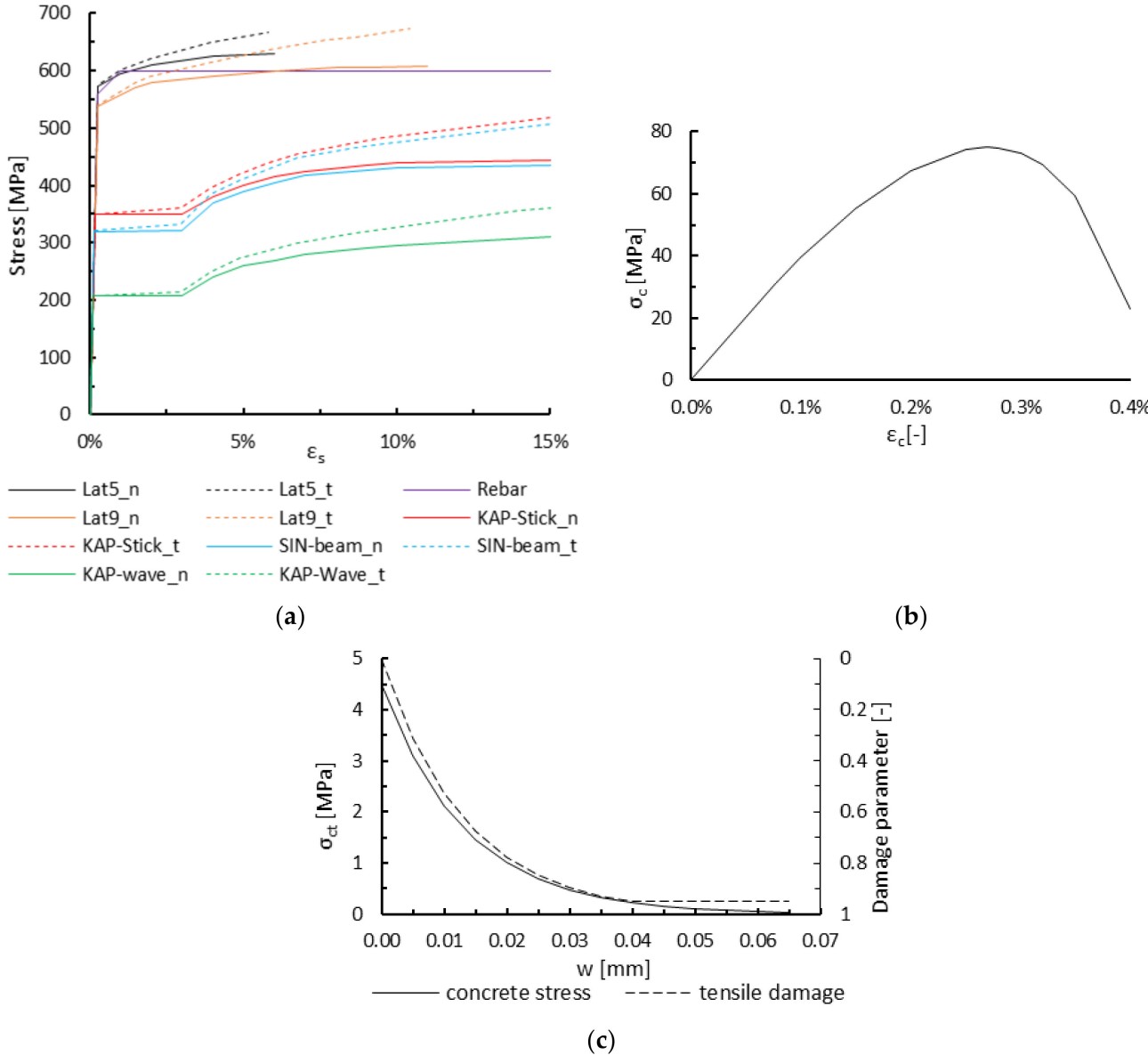

**Figure 6.** Material models used in the finite element software (**a**) steel materials with full lines for nominal stresses (n) and dashed lines for true stresses (t); (**b**) concrete compressive stress-strain relationship according to Sargin; (**c**) concrete tensile stress-crack opening relationship according to Hillerborg.

*2.2. Results*

The test results are displayed in Table 5 as ultimate loads and deformations as well as the respective shear force in the analysed specimen. The shear force $V_{Exp}$ was calculated according to (1) (see Figure 4). Furthermore, diagrams showing the applied load and the relative displacement, pictures of the laboratory specimen and finite element simulations post failure are shown.

$$V_{Exp} = P_{Exp} \times \frac{l_h}{l_v} \tag{1}$$

**Table 5.** Ultimate loads and deformations.

| Specimen | | $P_{max,exp}$ | $u_{P,max,exp}$ | $l_h$ | $l_v$ | $V_{exp}$ | $V_{mean,exp}$ | $P_{FEM}$ | $V_{FEM}$ | $V_{FEM}/V_{mean,exp}-1$ |
|---|---|---|---|---|---|---|---|---|---|---|
| | | [kN] | [mm] | [mm] | [mm] | [kN] | [kN] | [kN] | [kN] | [-] |
| Double wall elements | V1_Kap_1 | 9.89 | 4.02 | 326 | 400 | 8.06 | 7.96 ± 0.30 | 9.46 | 7.71 | −3.17% |
| | V1_Kap_2 | 10.06 | 3.74 | 326 | 400 | 8.20 | | | | |
| | V2_Kap_3 | 9.14 | 3.46 | 326 | 400 | 7.45 | | | | |
| | V2_Kap_4 | 9.98 | 3.10 | 326 | 400 | 8.13 | | | | |
| | V5_Lat5_1 | 3.80 | 1.71 | 326 | 400 | 3.10 | 3.53 ± 0.28 | 4.40 | 3.59 | 1.58% |
| | V5_Lat5_2 | 4.26 | 1.77 | 326 | 400 | 3.47 | | | | |
| | V6_Lat5_3 | 4.58 | 1.63 | 326 | 400 | 3.73 | | | | |
| | V6_Lat5_4 | 4.69 | 1.73 | 326 | 400 | 3.83 | | | | |
| | V7_Lat9_1 | 27.25 | 3.19 | 326 | 400 | 22.21 | 23.04 ± 1.00 | 26.50 | 21.60 | −6.26% |
| | V7_Lat9_2 | 27.66 | 2.60 | 326 | 400 | 22.55 | | | | |
| | V8_Lat9_3 | 27.81 | 3.60 | 326 | 400 | 22.67 | | | | |
| | V8_Lat9_4 | 30.36 | - | 326 | 400 | 24.75 | | | | |
| Thin panels with SIN-beams | V3_Welded_1 | 49.67 | 0.97 | 321 | 400 | 39.80 | 36.06 ± 3.74 | 42.36 | 33.94 | −5.87% |
| | V3_Welded_2 | 40.34 | 0.76 | 321 | 400 | 32.32 | | | | |
| | V4_transv_1 | 44.02 | 0.70 | 321 | 400 | 35.27 | 29.48 ± 5.78 | 23.85 | 19.11 | −35.19% |
| | V4_transv_2 | 29.58 | 1.38 | 321 | 400 | 23.70 | | | | |
| | V9_Flange supported_1 | 98.13 | 1.44 | 321 | 400 | 78.63 | 74.14 ± 4.49 | 101.35 | 81.21 | 9.54% |
| | V9_Flange supported_2 | 98.13 | 0.49 | 321 | 400 | 78.63 | | | | |
| | V10_Flange supported_3 | 86.92 | 1.56 | 321 | 400 | 69.64 | | | | |
| | V10_Flange supported_4 | 86.92 | 0.96 | 321 | 400 | 69.64 | | | | |
| | V11_Flange fixed_1 | 129.50 | 5.89 | 321 | 400 | 103.76 | 102.76 ± 1.00 | 119.77 | 95.97 | −6.61% |
| | V11_Flange fixed_2 | 129.50 | 5.22 | 321 | 400 | 103.76 | | | | |
| | V12_Flange fixed_3 | 127.00 | - | 321 | 400 | 101.76 | | | | |
| | V12_Flange fixed_4 | 127.00 | - | 321 | 400 | 101.76 | | | | |

*2.3. Discussion*

2.3.1. Double Wall Elements

KAP-Steel-Connectors

The load–deformation diagram in Figure 7a shows an almost linear behaviour in the laboratory tests until the KAP-sticks as well as the KAP-waves began to buckle.

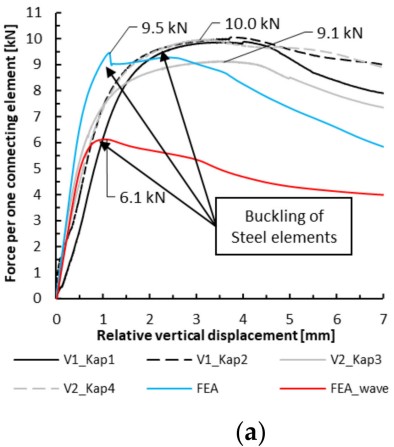
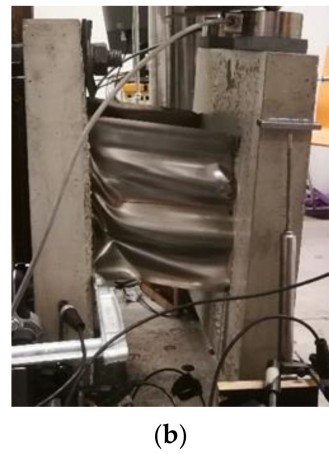
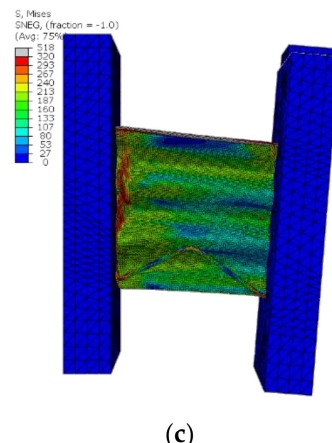

(**a**)　　　　　　　　　　(**b**)　　　　　　　　　　(**c**)

**Figure 7.** Test results KAP-steel-connector double wall elements: (**a**) load-deformation diagram with results from laboratory tests (black and grey) and FEA-simulations (blue and red); (**b**) specimen post failure in the laboratory test; (**c**) finite element specimen post failure, deformation superelevated two times.

The finite element analysis confirmed not only the ultimate load, but also showed, that the wave is responsible for a major part of the load bearing capacity. This can be seen by comparing the results curve from the model without sticks to the complete model. The load deformation diagram also shows that the finite element analysis slightly overestimates the stiffness from the laboratory tests.

The buckled KAP-wave and buckled KAP-sticks in the lower part of the specimen in Figure 7b,c after failure were the same for all specimens and did not cause any damage to the surrounding concrete.

Lattice Girders Ø5 mm and Ø9 mm Diagonals

The load deformation diagrams in Figure 8a,b show that until the buckling of the compressed diagonals, the behaviour was almost linear. In the diagram in Figure 8b, three of the four tested connecting elements are plotted due to problems at the deformation measurement of element V8_Lat9_4. As can be seen in Table 5, the load was in the same range as the other three lattice girders.

The results from the finite element parameter study (Figure 8a,b) on imperfections show the reduction in load bearing capacity and stiffness with increasing magnitude of imperfection. The best fit to the experiment was obtained at an imperfection of 4 mm for the 5 mm diagonals and at 5 mm for the 9 mm diagonals.

Table 2 shows the range of magnitude of imperfections, measured at two lattice girders. Superimposing deformations in the two directions shows that the results from the FE-simulations are plausible.

However, it must be considered that not all bars have the same pre-deformation and therefore behave more rigidly or softly, resulting in different stresses in the bars.

Figure 8c shows a 9 mm diagonals specimen after failure. Up to a deformation of approximately 10 mm, no damage was observed at the interface between the concrete and the lattice girder. After reaching the maximum load, the specimen was pushed further, deforming the already buckled steel bars, followed by a failure of the concrete panel (pull-out of the individual reinforcement bars of the lattice girder). This behaviour can be explained by a failure of the compressed bars, while the tensioned diagonals

were still under tension. The pull-out failure occurred at three of the four tested lattice girders with 9 mm diagonals and at very large deformations long after the ultimate load. This failure mode was also observed in the simulations for the 9 mm diagonals without imperfections (Figure 8b). At the 5 mm diagonals, no damage at the concrete next to the lattice girders could be observed, showing that the compressed bars have a fixed-fixed boundary condition.

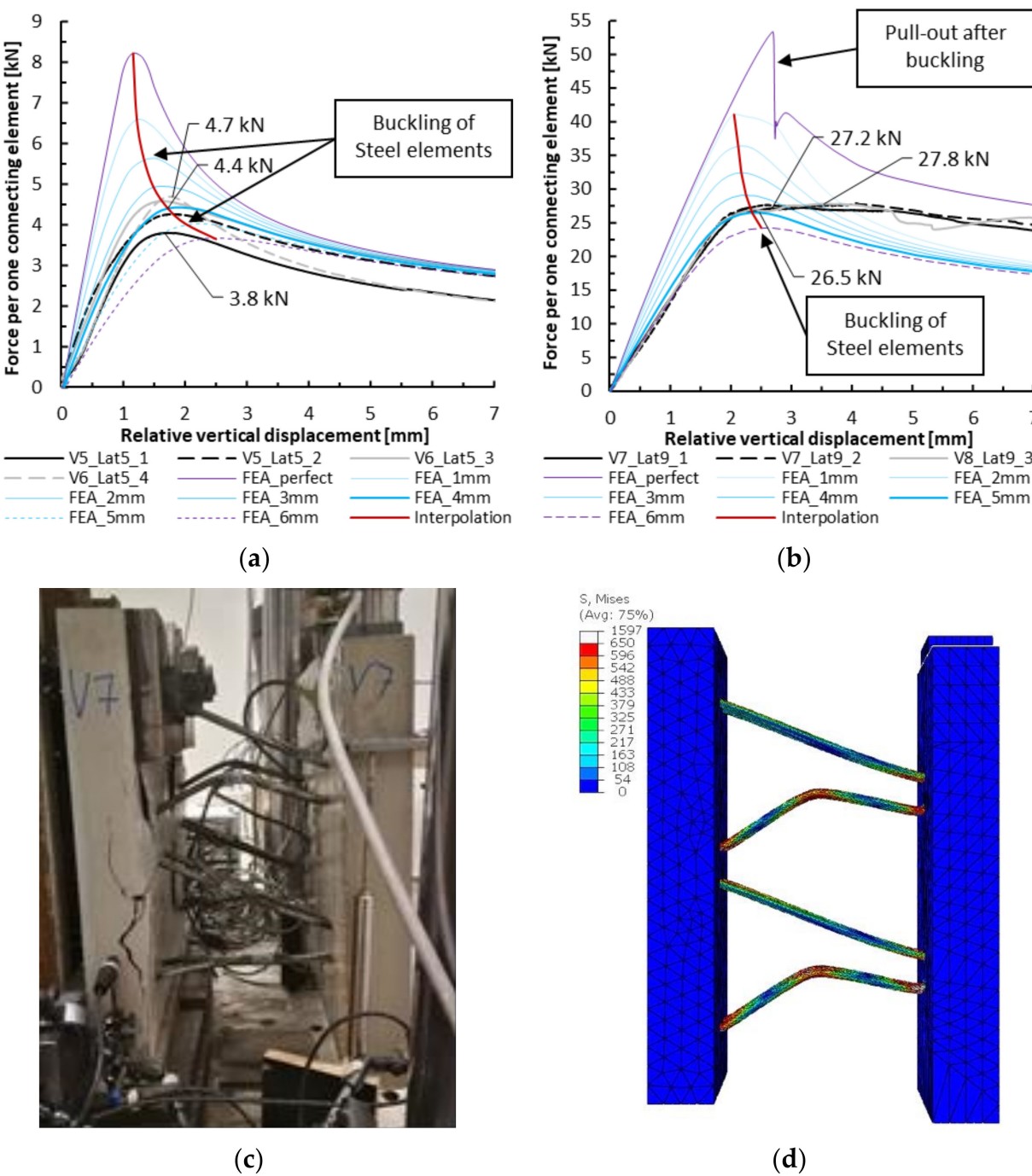

**Figure 8.** Test results Ø5 mm and Ø9 mm lattice girder double wall elements: (**a**) Ø5 mm lattice girder double wall elements: load-deformation diagram with results from laboratory tests (black and grey) and FEA-simulations (blue and purple); (**b**) Ø9 mm lattice girder double wall elements: load-deformation diagram with results from laboratory tests (black and grey) and FEA-simulations (blue and purple); (**c**) specimen post failure in the laboratory test; (**d**) finite element specimen post failure, deformation superelevated two times.

### 2.3.2. Concrete Panels with SIN Steel Beams
Version 1: Fixed Concrete Panel with Pull-Out of SIN Steel Beams

Both diagrams in Figure 9a,d show a sizeable difference in the ultimate load between the two beams in one test. At the version with the penetrating rebar, the stiffness was also significantly different.

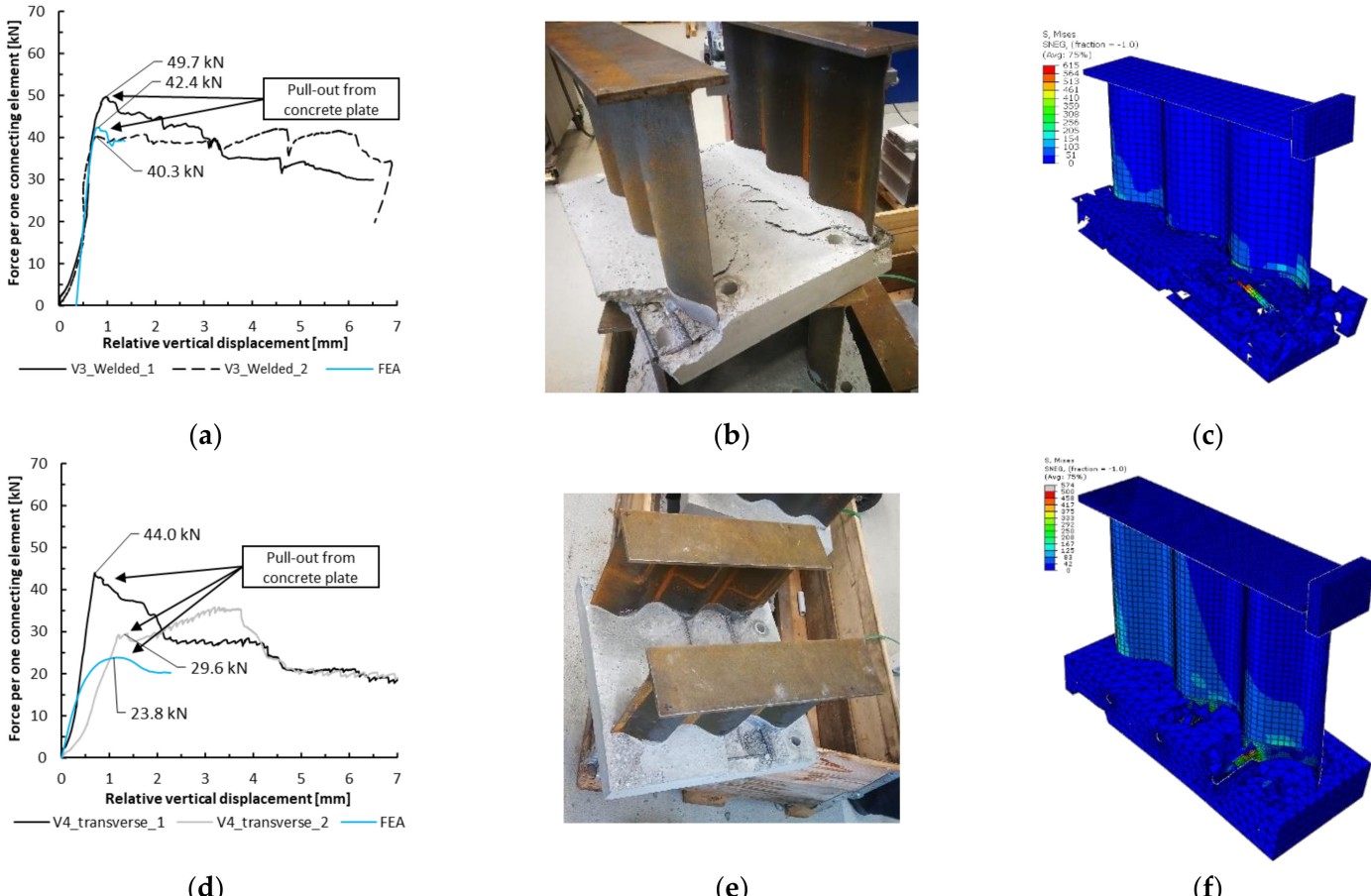

**Figure 9.** Test results: concrete panel and SIN beam, with fixed concrete panel (**a**) load-deformation diagram with results from laboratory tests (black and grey) and FEA-simulations (blue and purple); (**b**) specimen post failure in the laboratory test; (**c**) finite element specimen post failure, deformation superelevated two times (**d**) load-deformation diagram with results from laboratory tests (black and grey) and FEA-simulations (blue and purple); (**e**) specimen post failure in the laboratory test; (**f**) finite element specimen post failure, deformation superelevated ten times.

The specimen post failure with the pull-out of the steel beam from the concrete can be seen in (Figure 9b,e). Both failure modes were reproduced in the finite element simulation (Figure 9c,f), with a higher average load at the welded bars (Figure 9a,d). In the laboratory tests, an initial deformation was required so that all parts were under force, therefore the curve from the FEA was moved to the right in Figure 9a. The finite element simulation was stopped shortly after the maximum load due to huge computational effort without additional gain of information.

The ultimate loads obtained from the simulations fit to the experimental results for the welded version, while they were underestimated for the penetrated solution.

For a detailed understanding of the compound behaviour of corrugated webs with these systems, further investigations will be necessary.

Version 2: Upper Horizontal Support at the Steel Flange

Pullout was avoided with the new setup and no difference in the failure mode between the two specimen types could be observed. They failed due to local buckling of the corrugated steel web at the upper horizontal support (Figure 10b). The load–deformation diagram in Figure 10a shows that, after all components of the test setup were under force, approximately the same stiffness was found for all 4 steel beams. With respect to the maximum achieved test load, 86.7 and 98.1 kN were measured. The stiffness, ultimate load and failure mode were confirmed by the non-linear FE-simulation (Figure 10a–c).

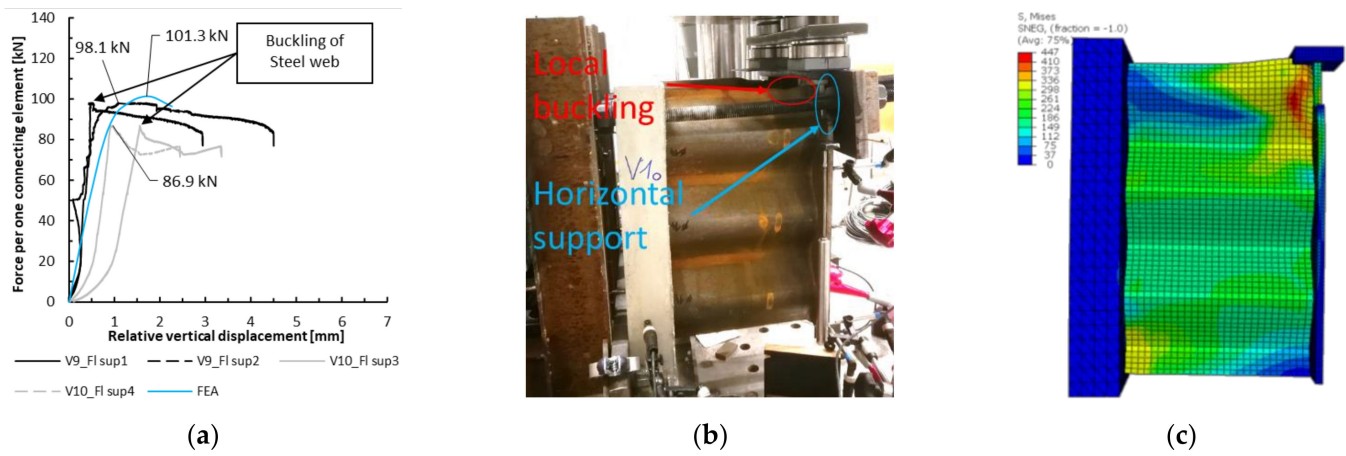

(a)          (b)          (c)

**Figure 10.** Test results: concrete panel and SIN beam, with horizontally supported steel flange (**a**) load-deformation diagram with results from laboratory tests (black and grey) and FE-simulations (blue); (**b**) specimen post failure in the laboratory test with local buckling of the steel web and horizontal support at the steel flange; (**c**) finite element specimen post failure, deformation superelevated two times.

Version 3: Steel Beam Fixed and Free-Floating Concrete Panel

The specimens from version 2 were tested again, with the steel beam as the fixed part and the concrete panel free-floating. Figure 11a shows the same maximum load for both girders. The test data for the second test are not displayed due to a loss of the experimental data. The maximum force is shown in Table 5. The specimen after failure in the laboratory in Figure 11b and the simulation in Figure 11c showed a local buckling of the corrugated web at the lower horizontal support. The difference in the stiffness is explained by the damage in the corrugated webs from the previous tests with the other setup.

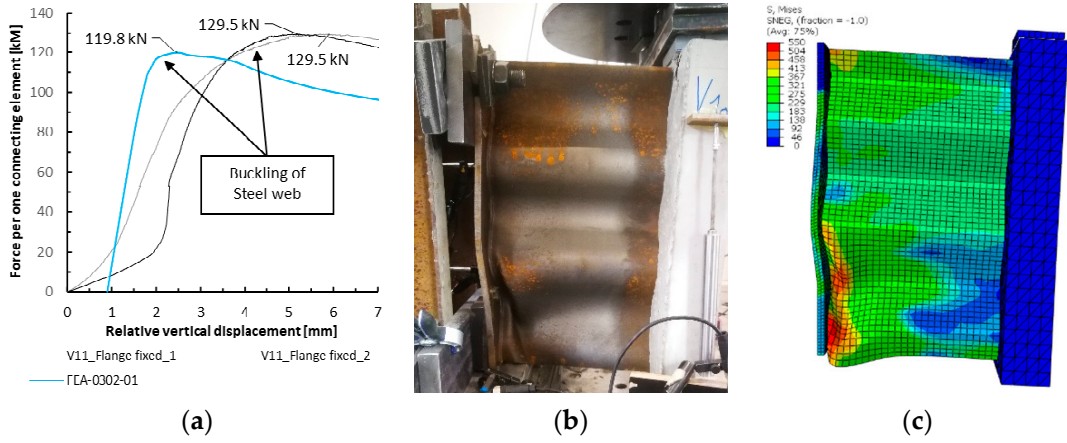

(a)          (b)          (c)

**Figure 11.** Test results: concrete panel and SIN beam, with fixed SIN beam (**a**) load-deformation diagram with results from laboratory tests (black and grey) and FEA-simulations (blue); (**b**) specimen post failure in the laboratory test with local buckling of the steel web; (**c**) finite element specimen post failure, deformation superelevated two times.

### 3. Push-Off Tests on Joints between Segments

*3.1. Test Design*

3.1.1. Specimen and Geometry

Push-off specimens similar to those used by other researchers [33,34] to study the shear transfer and aggregate interlock behaviour of pre-cracked concrete were adopted. Similar tests for dry and epoxied joints have been performed by several researchers [35–38]. The main research activities on the field of bridge segments with dry or epoxied connections are compiled in [39]. Further, grouted joints were also investigated in various experimental studies, primarily focusing on large precast panels' connection in building construction [40,41]. The specimens used in these tests consisted of pairs of concrete blocks (Figure 12) with a height of 260 mm and a width of 100 mm. These blocks were cast with a self-compacting concrete mixture with a target compression strength of 60 MPa, typical for precast elements. Four bent bars with a diameter of 10 mm were used to realize the corbels, which formed the load introduction points. Those were necessary to suppress bending effects in the tested joint.

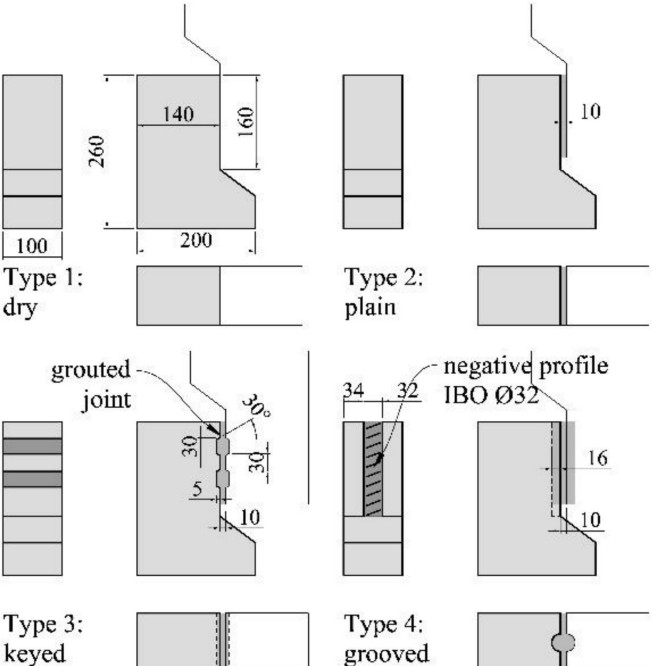

**Figure 12.** Geometry of push-off specimens with the joint geometry.

The connection between the separate blocks was achieved by a 10 mm thick layer of mortar, resulting in a joint area of $100 \times 140$ mm (types 2–4 in Figure 12). The layer thickness was identified as the minimum required tolerance in the construction of a full-scale prototype in [36]. Besides the connected specimens, a reference test without mortar was tested (type 1 in Figure 12).

Specimens of type 1 and 2 were plain joints with and without mortar. The surface was pre-treated with sandpaper (P40) for better adhesion. Specimens of type 3 had two shear keys with a depth of 5 mm and a width of 30 mm. The shear keys were designed to comply with the geometry rules of keyed surfaces in Eurocode 2, resulting in a total key surface $A_{Key}$ = 6000 mm$^2$. The shear transfer of type 4 was realised using a half-round vertical groove with a profiled surface, represented by a negative profile of a hollow-core drill IBO32 (rib height 1.6 mm and screw pitch of 12.7 mm, $A_{Groove}$ = 4480 mm$^2$). When two elements were put together, the two grooves formed a channel, which simplified the grout filling. This new joint-design was developed at the TU Wien, specially to address the proposed method in Section 1 and [36]. The surface of the specimens of type 3 and 4 had not been pre-treated before the grouting of the joints to spare this step with the

intention of saving construction time in the actual application. Since the separation along the groove in two specimens of type 3 occurred, an entirely untreated surface could not be recommended.

### 3.1.2. Materials

Two mortar products from two different suppliers were tested in this series, with the second grout only used on the specimen with joint type 2 and 4. Both products have expanding properties before setting and had low shrinkage properties. The maximum grain size was 1 mm for both mortars. Material properties were evaluated using mortar prism tests on the day of testing with dimensions of 40/40/160 mm (Table 6). A total of six compression tests and three flexural tests of each mortar were conducted. The material parameters of the concrete blocks were achieved by material tests on cylinders (d = 150 mm, h = 300 mm) consisting of three compression tests and three splitting tensile tests.

**Table 6.** Mechanical properties of mortar and concrete.

| Product | $\rho$ [g/cm$^3$] | $f_{cm}$ [N/mm$^2$] | $f_{ct,fl}$ [N/mm$^2$] | Age [days] |
|---------|-------------------|---------------------|------------------------|------------|
| Grout 1 | 2.24 ($\pm$ 0.7%) | 75.40 ($\pm$ 4.8%) | 10.30 ($\pm$ 4.7%) | 55 |
| Grout 2 | 2.26 ($\pm$ 0.3%) | 84.30 ($\pm$ 2.3%) | 6.00 ($\pm$ 5.6%) | 16 |
| Concrete | 2.33 ($\pm$ 0.7%) | 81.40 ($\pm$ 1.0%) | 4.10 ($\pm$ 1.3%) | 116 |

Both materials showed a very high concrete compression strength $f_{cm}$, while grout 2 showed a lower tensile strength $f_{ct,fl}$ during the flexural tests. Grout 2 was tested in a younger state than grout 1 considering the age of the specimens (Table 6).

### 3.1.3. Test Setup and Testing Procedure

Since main objective of these tests was to study the strength of the joints under uniform shear and normal stresses, a testing configuration was chosen that eliminated, to the extent possible, bending effects and stress concentrations (Figure 13).

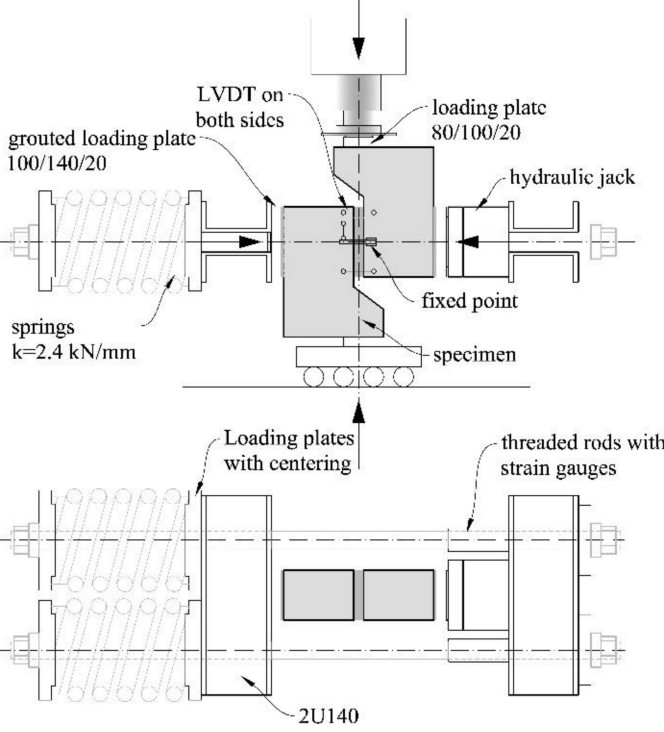

**Figure 13.** Test setup and surface measurements on push-off tests with constant lateral force.

Springs with a stiffness $k$ = 2.4 kN/mm were used to minimise a possible increase of the lateral force caused by dilatation of the joint.

Each conducted test consisted of the following steps: post-tensioning followed by a push-off test. The joint surface was prestressed with a nominal value of normal stress ($\sigma$ = 2.0/9.0/16.0 N/mm$^2$) in the joint. The post-tensioning forces $P$ were applied by a hydraulic jack over a frame containing two beams made of U-profiles and two threaded rods with strain gauges applied to them. The strain gauges were used to determine the prestressing force.

The applied load $F$ in the push-off tests was determined by measuring the oil pressure in the hydraulic jack. The dilatation $w$ of the joint was measured with a total of four LVDTs, placed on the top and bottom on each side. The shear deformation $s$ was measured with LVDTs placed in the middle on each side.

The experiments were loaded with a slow loading rate of 0.05 mm/min for precise coverage of damage effects. After the first pronounced load decrease, the loading rate was doubled. The test procedure was stopped after a shear deformation of at least 3.0 mm. Some experiments were continued until no further significant change in the post-peak behaviour was observed.

### 3.2. Results
#### 3.2.1. Load-Carrying Behaviour

The general characteristics of each joint type's bearing behaviour are plotted in Figure 14 and in order to explain the load-carrying behaviour. Figure 14 shows the dependences of the shear stress $\tau$ and coefficient of friction $\mu$ on the shear deformation s, respectively.

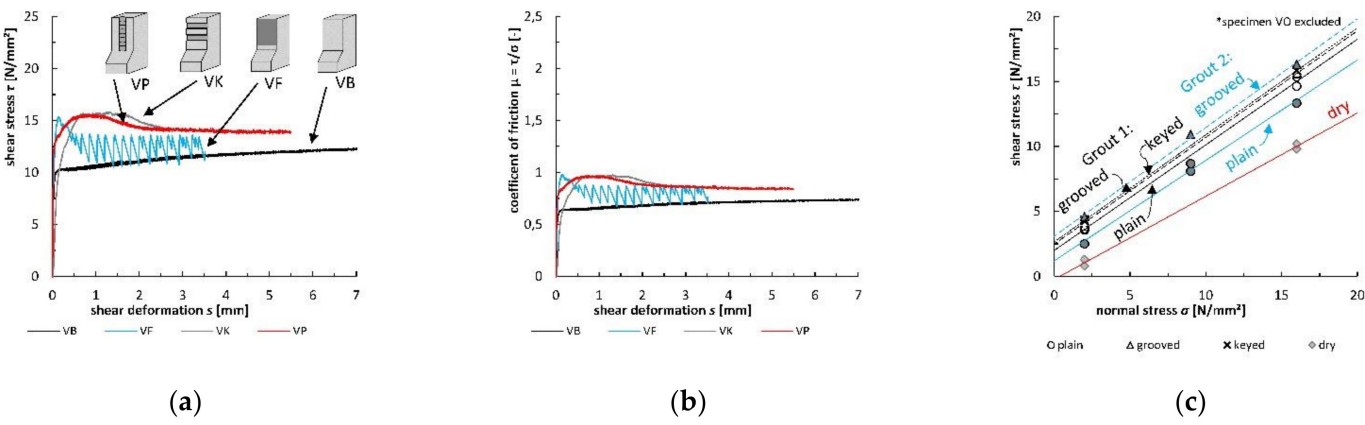

**Figure 14.** Load carrying characteristic of different joint types: (**a**) $\tau/s$–relationship; (**b**) $\mu/s$–relationship; (**c**) $\tau_{max}$ vs. $\sigma$.

#### Plain Grouted Joint

The plain grouted joints showed a different load-bearing behaviour when compared to dry joints. After reaching the maximum shear force, a significant load decrease occurred (Figure 14a). This difference between maximum load and the decreased load level afterwards characterises the adhesive component due to grouting. Subsequent to this first continuous decrease, the transferred shear force suddenly dropped to a dry joint load level but recovered afterwards. Although the same loading rate was used, the grouts showed a generally different behaviour (see VF in Figure 14a,b). The coefficient of friction $\mu$ was, as was the case for the dry joints, almost independent of the prestressing level but significantly higher at around 0.8.

#### Keyed Grouted Joint

In order to activate the full shear capacity of a keyed surface, a certain shear deformation was necessary, as shown in the load–deflection curve in Figure 14a. This failure

announcement behaviour is advantageous compared to plain joints, in which no significant movement occurs prior to failure. The keyed joints also showed a pronounced post-peak behaviour, since the failure load level could be held although shear displacement occurred. Only a minor force decrease addressed to an adhesive component could be identified. The friction coefficient for both prestressing levels converged at the end of the test and ranged between 0.8 and 1.0.

Grooved Grouted Joint

Similar to the keyed joint, a certain shear deformation was necessary to fully activate the shear capacity of the grooved joint. However, the load–deflection curve is different compared to the keyed joint since it shows an additional branch with a lower stiffness prior to maximum load level but at the same time stiffer at first. This failure behaviour could also be seen as advantageous in real-scale structures. The friction coefficient for all post-tensioning levels converged at the end of the test, lying between 0.8 and 0.9.

3.2.2. Influence of Investigated Parameters on Joint Capacity

In the diagram in Figure 14c, the maximum shear stress $\tau_{max}$ achieved in the experiments is plotted against the nominal prestressing stress $\sigma = P/A$, allowing a comparison of the resistance of the different joint types. All test results from this series are listed in Table 7.

**Table 7.** Test results and comparison to FEM analysis.

| Name | Grout | Type | $F_{max}$ [1] [kN] | $\tau_{max}$ [2] [N/mm$^2$] | $\sigma$ [N/mm$^2$] | $f_{cm}$ [N/mm$^2$] | $f_{ct,fl}$ [N/mm$^2$] | $F_{FEM}$ [kN] | $F_{max}/F_{FEM}$ [-] |
|------|-------|------|------|------|------|------|------|------|------|
| VA | - | 1 | 137.4 * | 9.8 | 16.0 | - | - | 155.1 | 0.89 |
| VB | - | 1 | 142.9 * | 10.2 | 16.0 | - | - | 155.1 | 0.92 |
| VC | - | 1 | 18.2 * | 1.3 | 2.0 | - | - | 19.4 | 0.94 |
| VD | - | 1 | 11.1 * | 0.8 | 2.0 | - | - | 19.4 | 0.57 |
| VE | 1 | 2 | 204.8 | 14.6 | 16.0 | 75.4 | 10.3 | 201.8 | 1.01 |
| VF | 1 | 2 | 215.0 | 15.4 | 16.0 | 75.4 | 10.3 | 201.8 | 1.07 |
| VG | 1 | 2 | 50.1 | 3.6 | 2.0 | 75.4 | 10.3 | 47.27 | 1.06 |
| VH | 1 | 2 | 51.4 | 3.7 | 2.0 | 75.4 | 10.3 | 47.27 | 1.09 |
| VI | 1 | 3 | 59.6 | 4.3 | 2.0 | 75.4 | 10.3 | 57.6 | 1.03 |
| VJ | 1 | 3 | 61.1 | 4.4 | 2.0 | 75.4 | 10.3 | 57.6 | 1.06 |
| VK | 1 | 3 | 221.8 | 15.8 | 16.0 | 75.4 | 10.3 | 229.7 | 0.97 |
| VL | 1 | 3 | 221.4 | 15.8 | 16.0 | 75.4 | 10.3 | 229.7 | 0.96 |
| VM | 1 | 4 | 54.5 | 3.9 | 2.0 | 75.4 | 10.3 | 54.0 | 1.01 |
| VN | 1 | 4 | 61.2 | 4.4 | 2.0 | 75.4 | 10.3 | 54.0 | 1.13 |
| VO | 1 | 4 | 187.4 | 13.4 | 16.0 | 75.4 | 10.3 | 211.0 | 0.89 |
| VP | 1 | 4 | 218.6 | 15.6 | 16.0 | 75.4 | 10.3 | 211.0 | 1.04 |
| VE2 | 2 | 2 | 186.5 | 13.3 | 16.0 | 84.3 | 6.0 | 185.0 | 1.01 |
| VF2 | 2 | 2 | 113.3 | 8.1 | 9.0 | 84.3 | 6.0 | 115.1 | 0.98 |
| VG2 | 2 | 2 | 121.6 | 8.7 | 9.0 | 84.3 | 6.0 | 115.1 | 1.06 |
| VH2 | 2 | 2 | 35.1 | 2.5 | 2.0 | 84.3 | 6.0 | 33.7 | 1.04 |
| VM2 | 2 | 4 | 228.4 | 16.3 | 16.0 | 84.3 | 6.0 | 211.0 | 1.08 |
| VN2 | 2 | 4 | 153.0 | 10.9 | 9.0 | 84.3 | 6.0 | 133.6 | 1.15 |
| VO2 | 2 | 4 | 64.6 | 4.6 | 2.0 | 84.3 | 6.0 | 46.7 | 1.38 |

\* $F_{0.2}$ at shear deformation of $s = 0.2$ mm is chosen for $F_{max}$. [1] experimental data of column $F_{max}$ also in [21]. [2] $\tau_{max} = F_{max}/A$ with $A = 14,000$ mm$^2$.

It became apparent that the relationship between the prestressing level and the maximum load $F$ could be described by a linear function in the investigated range. Therefore, only high and low prestress forces were investigated in the grout 1 series.

The positive effect of joint grouting on the shear resistance can be seen when the results of specimens with plain joints are compared to the results with dry joints (light grey line in Figure 14c). The plain grout 1 specimens showed an average increase of 53 kN, while the increase with grout 2 specimen was about 40 kN.

The comparison of grooved joints with plain joints shows that grooved joints have a higher shear capacity, presumably because of their higher roughness. Nevertheless, the increase is more pronounced with the grout 2 specimens, seeing an average increase of 36 kN. By contrast, only a small increase of about 8 kN could be observed in the Grout 1 specimens. It is assumed that the different ratio between the compressive and tensile strength of the mortars leads to this difference.

The keyed joint showed almost the same load capacity as the grooved joint (see Table 7). However, as discussed above, the load deflection curve is different.

### 3.3. Discussion of Failure Cause by Non-Linear FEM-Analysis

The non-linear FEM analysis was performed using ABAQUS FEA with the previously introduced material and parameters (Table 3). The interface parameters were calibrated by the test results of the plain grouted joint. The simulations can fully reproduce the achieved test loads (Table 7). Moreover, the load carrying behaviour is represented quite accurately with some lack in the post-peak behaviour of the keyed and grooved joints (Figure 15). Pronounced plateaus at load-peak were observed in the experiments, which could not be covered in the simulations. These plateaus could be explained by delamination processes of these rough interfaces, accompanied by an early shear deformation, both of which were observed in the experiments. These computing time intensive delamination processes were not integrated in the FE-model of this investigation. The comparison between the cracking pattern of the tests and the simulations is shown in Figure 16.

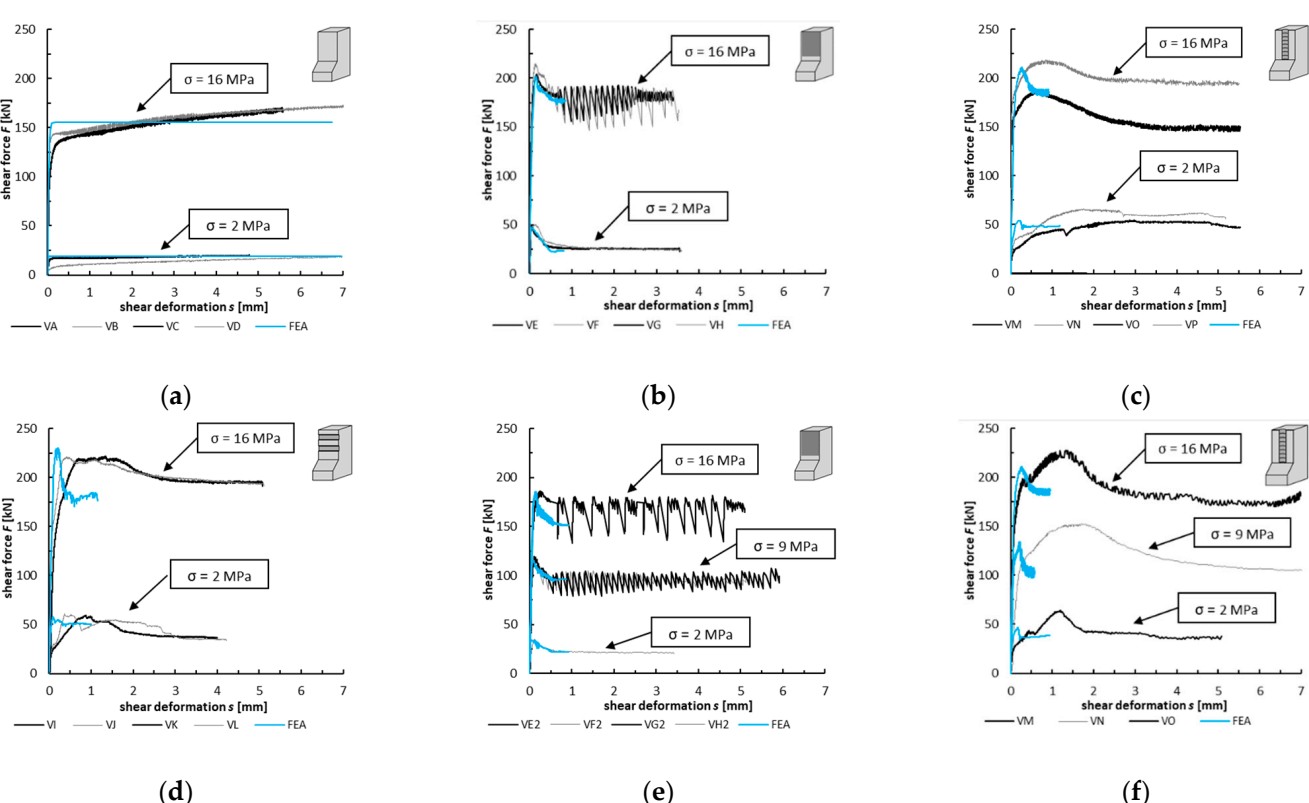

**Figure 15.** Load-deformation diagrams compared to simulation results for (**a**) dry joint; grout 1: (**b**) plain; (**c**) groove; (**d**) keyed and for grout 2: (**e**) plain; (**f**) grooved, respectively.

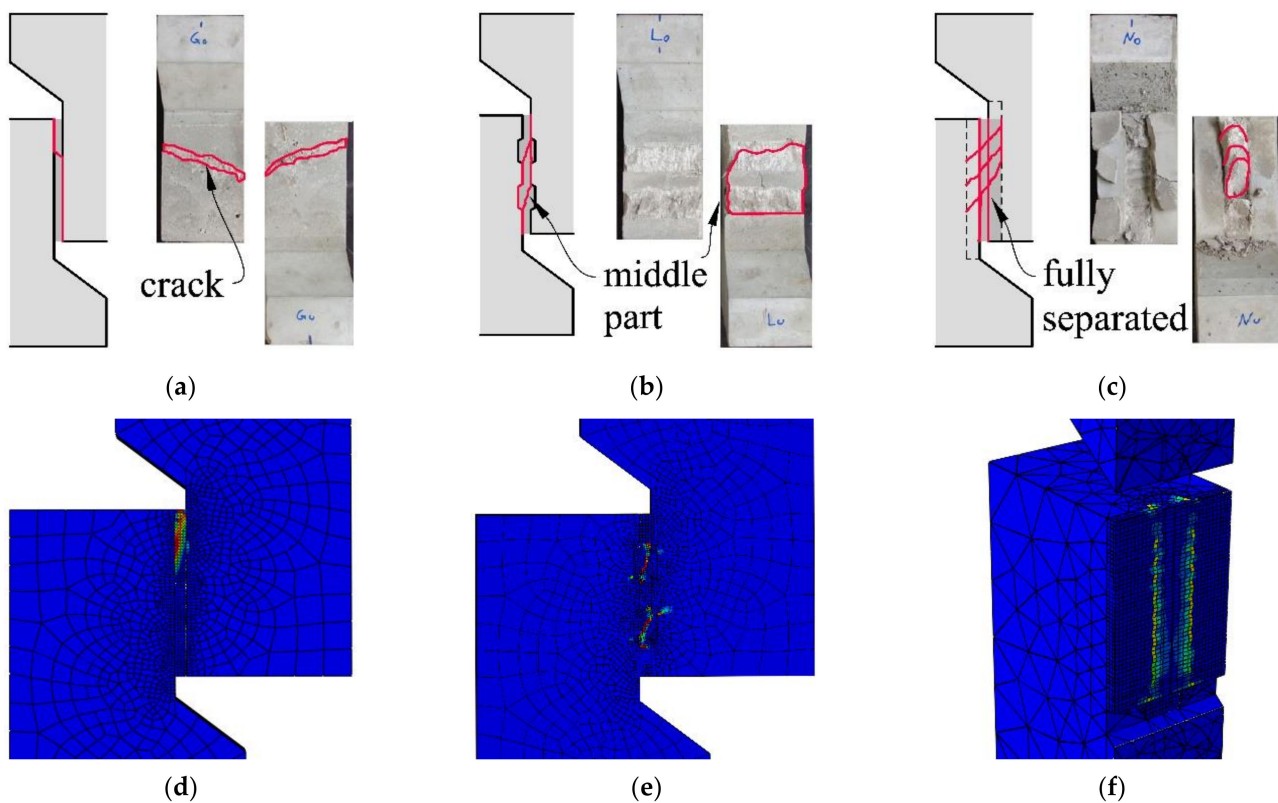

**Figure 16.** Typical cracking pattern for different joint types: (**a**) VG (plain); (**b**) VL (keyed); (**c**) VN (grooved). Comparison to the FEM results: (**d**) plain, (**e**) keyed, (**f**) grooved.

The failure in the plain grouted joint is defined by overcoming the frictional capacity plus an adhesive portion according to the shear-friction model of the fib Model Code 2010 and Eurocode 2. After reaching this load level, delamination occurs, accompanied by shear deformation, which leads to a single inclined crack in the grout located approximately at the middle of the joint. In the end, the grout is fully separated, with the rest of the material still attached to the opposite concrete blocks in simulations and testing (Figure 16a). The bearable load drops to the level of the test with dry surface, which could be explained by the fact that the grout slides as a quasi-negative profile on the dry surface. The simulations also showed this failure surface.

After overcoming this preliminary shear friction, the shear displacement leads to an activation of shear keys at the specimen with the keyed surface. When this happens, the formation of concrete compression struts occurs (Figure 16b). The load increase is then limited by a diagonal cracking initiated at the contact point of the upper shear key on the specimen half with load introduction (Figure 16b). The same crack can be found on the lower shear key on the specimen half with the load-bearing point. Finally, the grout part is also separated in the middle in tests and simulations.

The grooved joint can be interpreted as a rougher surface that acts along the plain surface. Thus, the bearing portions of both surfaces could be aggregated. Until this load level is reached, only a slight shear deformation occurs (Figure 14a). When this load level is surpassed, the separation between concrete block and grout leads to a grind down of the ribs (Figure 16c) and also to a separation of this grouted pillar and the plain grout at various locations in the joint. Only in test VO was a different cracking pattern with an almost linear separation of the pillar observed, resulting in a lower transferable force (Table 7).

## 4. Conclusions

In this paper, shear tests on different connecting elements for unfilled double walls (lattice girders with Ø5 mm and Ø9 mm diagonals and KAP-steel-connectors) as well as thin-panels with attached steel girders with corrugated webs were presented. The load bearing behaviour and the failure modes were analysed as such elements will be utilised for the creation of cross-frames in bridge segments made from thin-walled pre-fabricated elements.

Further, different joint designs for thin-walled pre-fabricated elements for the application in segmental bridge construction were discussed. The load-bearing behaviour of those joints was analysed by push-off tests with a constant lateral force. The main parameters are the joint type (wet joints: plain, grooved, keyed; dry joints), the grout type and the level of lateral force.

The test results of both investigations were evaluated by non-linear finite-element simulations using the commercial software ABAQUS. Based on the conducted evaluations following conclusions are drawn.

- For the prototype bridge segment shown in Figure 1, the maximum shear force in the vertical webs is 16.81 kN [42]. When comparing this to the shear capacities of the tests (Table 5), it can be seen that this load can be bourn by all variants of steel beams and the lattice girders with Ø9 mm diagonals. For the thinner lattice girders, it is possible to create a frame by using five girders instead of one, while for the KAP-steel connectors three rows would be necessary. The use of multiple girders in a double wall can be advantageous when it comes to filling the double wall with concrete. A higher number of connections between the two shells allows for a higher filling speed. The tests also showed that the lattice girders diagonals of unfilled double wall elements can be considered as fixed inside the concrete; this also applies to the KAP-steel-connectors.
- All experiments could be reproduced with the finite element simulations, helping to better understand the different failure modes. Parameter studies on the imperfections of lattice girders showed that such slender compressed struts are very sensitive to imperfections as the stiffness and the load bearing capacity are reduced with the magnitude of imperfection.
- Nevertheless, for the assessment of the corrugated webs additional investigations on the compound behavior and the shear capacity are necessary, especially when it comes to girders with holes for post-tensioning ducts.
- Grouted joints are suitable for connecting bridge segments made of pre-fabricated thin-walled elements since a continuous load transfer could be enabled during the erection process. In addition, the closed-shell formed thereby serves as formwork in the later construction process.
- The newly introduced concept of profiled grooved joints enables an easy grouting process and therefore enables smaller joints. The test results show that this joint type is as capable of transferring shear forces as joints with a typical shear keyed surface. Both types display a ductile behaviour that is favourable in comparison to the grouted plain joint.
- The conducted FEM-simulations are capable of reproducing the results of the small-scale tests presented. Further, parameter studies on the effect of the mechanical properties of the grouts and the redistribution of prestressing forces in the full-scale model will be conducted.

**Author Contributions:** Shear tests on cross-frames, test conceptualization, S.J.F. and J.K.; methodology, S.J.F.; investigation, S.J.F., M.R. and T.H.; formal analysis (FEA), S.J.F., writing, S.J.F.; supervision, J.K.; Push-off tests, test conceptualization, T.H. and J.K.; methodology, T.H.; laboratory tests, T.H. and S.J.F.; formal analysis (FEA), M.R.; writing, T.H.; supervision, J.K. All authors have read and agreed to the published version of the manuscript.

**Funding:** This research was funded by FFG, grant number 880272. Open Access Funding by TU Wien.

**Institutional Review Board Statement:** Not applicable.

**Informed Consent Statement:** Not applicable.

**Data Availability Statement:** The data that support the findings of this study are available from the corresponding author upon reasonable request.

**Acknowledgments:** The research project "Bridge construction with thin-walled segments out of pre-fabricated elements" is organised by the Austrian Society for Construction Technology (ÖBV) and financially supported by the Austrian Research Promotion Agency (FFG). as well as by the following companies: ÖBB, ASFINAG, Porr, Strabag, Swietelsky, Habau, Implenia, Hochtief, Zeman, Östu-Stettin, Leyrer & Graf, Oberndorfer, ANP-Systems, VÖB, VÖZ, FCP, Baucon, Schimetta, Öhlinger & Partner, ZT Mayer. The authors express their sincere gratitude for the financial support. The lattice girders and the SIN beams were provided by the companies Filigran and Zeman, the authors thank them for this support. The authors acknowledge TU Wien Bibliothek for financial support for editing/proofreading.

**Conflicts of Interest:** The authors declare no conflict of interest.

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
