# Peer review of "Semi-Precast Segmental Bridge Construction Method: Experimental Investigation on the Shear Transfer in Longitudinal and Transverse Direction"

_applsci, doi:10.3390/app11125502_

Round 1

Reviewer 1 Report

Research well prepared and technically carried out. The research results have been correctly presented and therefore are clear and understandable.

Author Response

Thank you for taking the time to review our article and for your feedback!

Reviewer 2 Report

Figure 14.  - the picture is not in colors, so it's quite difficult to find VB, VF, VK, VP - just two grey lines and two black lines

Line no.441 "....different behaviour (see VF in Figure 14a, b)"  - as mentioned above is not possible to find VF

in the table 7 - Test results; experimental data reported in [21], the data presented in the column  "Name"  should be explained more detail - not everybody can find results presented in Proceedings of IABSE Congress Christchurch 2020.

Author Response

Thank you very much, for taking the time to review our article and for your valuable comments.

We have taken them all into account as they really improve the quality of our article.

Changes in the manuscript are highlighted in yellow.

Fig 14: "Figure 14 was changed to colours for better readability as requested by the reviewers"

Table 7: " The authors appreciate this remark. The intention of the authors was to clearly state, that the experimental data achieved by the push-off tests could also be found in a conference paper of the authors. The description of the experiments and the methodology is extensively described in the current paper. Further footnotes were added and the table header was changed."

Reviewer 3 Report

The paper “Semi-precast segmental bridge construction method:

Experimental investigation on the shear transfer in longitudinal and transverse direction” investigated the shear transfer in precast box girders in different directions. The load capacity of different shear transmitting elements has been evaluated. The research sounds very interesting in terms of the topic and applicability to the field of bridge engineering. The introduction and the methodology have been explained in a fair manner. However, I recommend the authors to revisit the results and discussion and the summary/conclusion parts. In my opinion, such an important research can be better organized in terms if results justification. A comprehensive comparison between different shear transferring elements has not been introduced in the discussion, nor the summary and conclusion parts.

It is also not clear that the loads considered for the analysis includes the live load or no? If yes, has any provision been considered for fatigue issue?

I highly recommend the authors to revise the paper organization to make the research more applicable for bridge engineers and researchers. Provide a shorter table of comparison between different shear transfer elements and provide statistics on graphs and results.

Author Response

Thank you very much for taking the time to review our article, we really appreciate your input!

We copied your text and added our answers in green colour at the respective position.

Experimental investigation on the shear transfer in longitudinal and transverse direction” investigated the shear transfer in precast box girders in different directions. The load capacity of different shear transmitting elements has been evaluated. The research sounds very interesting in terms of the topic and applicability to the field of bridge engineering. The introduction and the methodology have been explained in a fair manner. It will remain as it is.
However, I recommend the authors to revisit the results and discussion and the summary/conclusion parts. In chapter 2.2 the results (ultimate loads and deformations) are presented in a table and discussed in chapter 2.3 for each type of element (including detailed load deflection diagrams). We would like to keep those parts as they are.

In my opinion, such an important research can be better organized in terms if results justification. A comprehensive comparison between different shear transferring elements has not been introduced in the discussion, nor the summary and conclusion parts. Table 5 shows the results of all tested connecting elements, also with the mean value for each type of connecting element. In Chapter 4 at the first point of the list of conclusions, the different connecting elements are compared with a reference to table 5. We would leave this section as it is.
If you require changes, we would be grateful if you could provide us more detailed input.

It is also not clear that the loads considered for the analysis includes the live load or no? If yes, has any provision been considered for fatigue issue?

Figure 2 shows the construction process. The first sentence after figure 2 describes that this research is about the shear transfer during construction stages. Therefore liveloads are only considered during construction and fatigue is not relevant for the analyzed load cases and constructions.
Therefore we would like to leave this as it is.

I highly recommend the authors to revise the paper organization to make the research more applicable for bridge engineers and researchers.

The other two reviewers gave us positive feedback on the paper organization. We discussed your recommendation and came to the results that we would like to leave the paper organization as it is. We hope for your understanding

Provide a shorter table of comparison between different shear transfer elements and provide statistics on graphs and results.

We discussed the option of creating another shorter table. Due to the fact that we would repeat the results from table 5, we decided not to do this. An option would be to highlight the mean values of the results in table 5.

Once again thank you very much for your critical review and the feedback you gave us. We really discussed your input a lot and value it very much!

Round 2

Reviewer 3 Report

Thanks for the response to the comments.